# From "Sangha Forest" (叢林 Conglin) to "Buddhist Academy": The Influence of Western Knowledge Paradigm on the Chinese Sangha Education in Modern Times

**Yifeng Liu**

School of History (Institute for Global History), Beijing Foreign Studies University, Beijing 100089, China; liuyifeng@bfsu.edu.cn

**Abstract:** Drawing on Foucault's theoretical framework of "space and power", this paper examines the discursive construction of "knowledge" in the context of Chinese Buddhist education. It traces the historical transformation of Chinese Buddhist education from the traditional "Sangha Forest"(the monastic community; 叢林 Conglin) style education to the Buddhist Academy, and analyzes how modern Buddhism reshaped its social image and function from a faith-based to a knowledge-based culture. Furthermore, this paper explores the reasons why modern Buddhism requires "knowledge" as a bridge between its worldly and transcendental dimensions, and the roles of elite laymen and monasteries as "Buddhist Institutes" in the new discursive practice.

**Keywords:** modern Chinese Buddhism; Sangha Education; Buddhist Academies; Buddhist institutes; Buddhist monastic space

## 1. The Internal and External Constraints Imposed by the Discursive Power of "Knowledge" on Chinese Buddhist Education

*1.1. External Constraints: Traditional Chinese Epistemology Reshaped by the Modern Western Education System*

From 1850 to 1949, China experienced a critical period of modern education and scientific transformation. The reformists of the late Qing dynasty believed that to revitalize China, they had to start by reforming the people's mentality and the education system. In 1862, the Qing government established the Tongwen Guan of Peking (京師同文館 Jingshi Tongwen Guan) as the earliest modern school in China, marking the beginning of modern education in China. In 1898, the Qing government founded the Imperial University of Peking (京師大學堂 Jingshi Daxuetang) as the first national comprehensive university in modern China. In 1905, the Qing government abolished the imperial examination system that had lasted for more than a thousand years and set up the Ministry of Education (學部 Xuebu) as the central agency for education. It also issued *the Imperially Approved Regulations for Schools* (钦定學堂章程 Qinding Xuetang Zhangcheng) as the first official school system promulgated by the government and implemented nationwide. After that, the Qing government established various types of new schools and specialized institutions, such as the Foochow Arsenal Academy (福建船政學堂 Fujian Chuanzheng Xuetang), the Higher Normal School (高等師範學堂 Gaodeng Shifan Xuetang), and the School of Practical Industries (實業學堂 Shiye Xuetang). It also sent students to study abroad in Japan and Europe. From the Tongzhi Restoration (同光新政 Tongguang Xinzheng), the Self-Strengthening Movement (洋務運動 Yangwu Yundong), the Hundred Days' Reform (維新變法 Weixin Bianfa), the New Policies of the Late Qing (清末新政 Qingmo Xinzheng), the Constitutional Preparation Period (預備立憲 Yubei Lixian) to the Beiyang government (北洋政府 Beiyang Zhengfu) and the Nanjing government (南京政府 Nanjing Zhengfu) era, Chinese intellectuals learned technology and science from the West in various ways. The education model centered on Western learning not only changed the intellectual, intuitive,



and individual modes of traditional Chinese education, but also introduced a knowledge system that was essentially different from Eastern civilization's "science". Science, as a new type of knowledge system, began to spread and popularize in China, especially after the May Fourth Movement. Science was no longer just about Western technology such as making guns, ships, and steel, but a rational tool that could explain the world, transform society, and improve individuals. The dissemination and popularization of scientific knowledge also brought a new form of power, that is, the power to shape people's bodies and minds through various mechanisms such as knowledge, discipline, surveillance, etc., thereby producing obedient and useful subjects. This view was profoundly discussed by French sociologist Michel Foucault in his book *The Birth of The Prison* (Michel Foucault 1977, pp. 135–69). He believed that there was a positive feedback loop between the formation of knowledge and the expansion of power. Some forms of knowledge can be dissolved by the intervention of power relations, while power itself can be amplified by the formation and accumulation of new types of knowledge (Michel Foucault 1977, pp. 27–31). Therefore, Western learning as a kind of knowledge also represents a new force that undermines the traditional Chinese education system.

This emerging force not only had a profound impact on China's secular education, but also brought unprecedented challenges to China's religious education. The great changes in modern Chinese education from concepts to systems also impacted Buddhist education. Modern Buddhist education not only had to face the competition and conflict of the new education model, but also had to accept the infiltration and transformation of the new knowledge system and value system. This new knowledge system and value system mainly came from Western modernist thought, which had a close connection with anti-clericalism. Vincent Goossaert and David Palmer talked about this in their book *The Religious Question in Modern China*, saying that after the Hundred Days' Reform (百日维新 Bairi Weixin) of 1898, Western modernist thought poured into China, and influenced by Western religious concepts, Chinese intellectuals began to re-examine Chinese religion in the framework of "superstition and religion" (Vincent and Palmer 2011, p. 91). The Ministry of Internal Affairs introduced the *Regulations for the Management of Buddhist Temples and Monasteries* (《管理寺廟條例》 Guanli Simiao Tiaoli), which stipulated in Article 5 that all temples and self-established schools must teach general education in addition to Buddhist scriptures. Article 15 of the *Regulations* also made specific provisions on the teaching objectives, methods, and content of the temple schools, which marked that introducing a modernized new education model into Buddhist education was not only the aspiration of the people, but also a systemic requirement. At the same time, there were also popular movements such as destroying superstition, transforming temple property into schools, etc., which reflected a tendency to challenge the authority and legitimacy of traditional religious institutions and practices and to advocate scientific and rational ways of understanding reality.

The pressure of social reality made the monks gradually see the precariousness of Buddhism's position. The Education Initiation with Temple Property Movement (廟產興學運動 Miaochan Xingxue Yundong) further prompted Buddhist practitioners to reflect on the many problems that had developed within the Buddhist community. This reflection then paved the way for the introduction of a new monastic education system. Given the twists and turns the community had experienced since the Hundred Days' Reform until the recent movement to expropriate monastic properties for making schools, it started to conduct critical self-reflection, during which community members identified the lack of well-trained, high-quality Buddhist practitioners as the key reason for the decline of Buddhism. Thus, they concluded that "the only way to save monastic properties from devastation is to quickly install a Buddhist education system 果欲維护寺产，避免遭受摧残，唯有火速兴办教育事业". In response to the crisis, Chinese Buddhism, in agony, attempted to preserve its own sacred space by developing a modern monastic education system. Buddhist institutes were soon established all over the country. Without the example set by Western missionary schools, the transition to modern Chinese education, and the recent expropria-

tion of monastic properties for building schools, the Buddhist community probably would not have recognized the urgency with which it needed to develop its own education system. It is precisely for this reason that Dongchu 東初 (1908–1977) proposes the year 1898, the 24th year of the Guangxu Era and the year of the Hundred Days' Reform, to mark the beginning of modern monastic education (Dongchu 1974, p. 203).

*1.2. Internal Constraints: The Decline of Anti-Intellectual Monastic Education in Traditional "Sangha Forest" (叢林 Conglin) in Modern China*

In Buddhism, the importance of knowledge is to be realized within the framework of "right faith" (正信 Zhengxin)-"superstition" (迷信 Mixin). The traditional Buddhist education has continued the context of "respecting intelligence" from the Era of the Dharma Commentator (論師時代 lun shi shi dai) in the Wei, Jin, Southern, and Northern Dynasties, to the Era of the Dharma Masters (法師時代 Fashi Shidai) in the Sui and Tang Dynasties. The "intelligence" here emphasizes the dialectics of Buddhist doctrines. During the Wei, Jin, Southern, and Northern Dynasties, a large number of Buddhist scriptures were introduced into China, and monks studied, explained, and debated them. During the Sui and Tang Dynasties, monks translated and annotated the Buddhist scriptures and exchanged and compared them with Indian Buddhism, forming various sects. Buddhist monks improved their wisdom and insight in the atmosphere of doctrinal dialectics, and Buddhist education mainly unfolded through lecturing on scriptures and dharma and discussing principles and meanings. Learned monks made Buddhism respected and revered in society at that time. After the late Tang Dynasty, with the rise of Chan Buddhism, Buddhist education took another path of "anti-intellectualism". Chan practice is to attain enlightenment through intuitive ways of spiritual experience such as "observing heart" (觀心 Guanxin), "seeing heart" (看心 Kanxin), "sealing heart" (印心 Yinxin), etc., and considers knowledge and words to be limited, thus neglecting the training of cognitive awareness. Moreover, since the Song Dynasty, Chan's sense of dharma lineage was influenced by the cultural gene of Chinese society that centered on bloodline and clan system, emphasizing the teacher-disciple relationship and the transmission of orthodoxy. "It highlighted the authority of the patriarchs in the process of transmission and promoted the formation of patriarchal faith (Fang 2012, p. 805)." Therefore, after the Buddha-dharma, as the "truth" changed from being carried by texts to being carried by individuals (patriarchs), Buddhist education tended to deepen the clan concept, gradually eroding the purity of Buddha-dharma faith. However, this situation changed following the rise of Chan Buddhism during the later years of the Tang dynasty, when "anti-intellectualization" became the new norm dominating the developmental path of Chinese Buddhism. Since the Song dynasty, in integrating the key concepts of patriarchy, Chinese Buddhism has established lineages to assert and normalize the spiritual "bloodlines" passed down from teachers. Subject to the influence of notions such as bloodlines and patriarchal hierarchies that characterize Chinese culture and society, the concept of Dharma in Chan Buddhism "emphasizes the authority of the teachers during the process of Dharma transmission and thus facilitates the development of students' faith in their teachers (Fang 2012, p. 805)."

This anti-intellectual tradition in Chan Buddhism subsequently caused malpractices to occur, while the truth of the Dharma was gradually lost over time. As the ancient saying goes: "An object that attracts insects must have already gone rotten, a body that attracts diseases must have already been weak and unhealthy 物必自腐而後蟲生，身必自虛而後病入". Considering the quality of Buddhist education in modern China, not only was there a significant gap between Buddhist teachings and the social realities at the time, but the quality of the teachers was also lamentable. The lack of good teachers, the self-contained manner characterizing Buddhist education, the preference to employ only one's close relatives or friends, and pay inequality are among the factors that led to the decline of the traditional "Sangha Forest" (叢林 Conglin; hereafter referred to as the Conglin) education and the drastic reduction in teaching quality. This change was particularly the case with subjects that emphasized the practice of faith, such as Chan Zong (Chan Buddhism 禪宗), Jiaozong

(teaching the sutras 教宗), and Lüzong (Rissū Buddhism 律宗). Consequently, as Gong Zizhen 龔自珍 (1792–1841) observes, "With its dropped standards, now it seems that anyone can practice Chan Buddhism, which, in turn, causes this Buddhist school to further lower its threshold and its doctrines to become even simpler. The literati are content with this easy access to the Chan school, as they can now justify nearly all their behavior in the name of Chan. Meanwhile, the illiterate monks are also practicing Chan in a shallow and frivolous manner. The number of available Chan booklets now even exceeds that of novels. Some insane people gathered opera singers, taught them Chan using simple phrases improvised on the spot, and asked them to sing these with good rhythms and tones during their performances. Then, three days later, there were Chan masters everywhere! 愈降愈濫，愈誕愈易。不但昧禪之行、冒禪之名儒流文士樂其簡便，不識字髡徒習其狂滑。語錄繁興，多於小說……狂者召伶市兒，用現成語句授之，勿失腔節。三日，禪師其遍市矣！". According to Yinshun 印順 (1906–2005), "the decline of Chinese Buddhism happened not only because of its vagueness and shallowness and the lack of critical thinking by its practitioners but also because the Buddhist community tended to focus on mystical theories at the expense of ignoring facts. Ever since the Song dynasty, one cannot easily find any satisfying biographies of senior monks. This situation lasted until then when the contempt of knowledge and dislike of critical theories finally led Chinese Buddhism into complete chaos." As traditional monastic education lost its original quality, previously marginalized phenomena such as superstition and the vulgarization of Buddhism thus became severe problems.

In modern China, the key to revitalizing Chinese Buddhism was thus reviving its earlier tradition of "respecting intelligence". As for the young generation of Buddhist practitioners who had grown up in the new era, "their behavior was subject to the regulation of self-discipline rather than traditional discipline, while their faith was developed based on a rational understanding of Buddhist doctrines rather than superstition (Deng 1994, p. 146)." Therefore, in the overall history of Buddhism, the promotion of modern Buddhist institutes can be seen as a restoration of the Dharma to Chinese Buddhism. Although traditionally, Buddhist practitioners in favor of Conglin Style education tended to be reluctant toward knowledge acquisition, it is undeniable that, since the commencement of the modern era, the discursive power of knowledge has still left its mark on Conglin practitioners.

## 2. The Rise of Knowledgeable Laymen

### 2.1. Elite Laymen and Their Initiative in Buddhist Studies

Since the 19th century, European religious studies have moved from a traditional theological background to an intellectual path. The so-called intellectualization means to analyze and explain religious phenomena with scientific methods and theories rather than relying on divine revelation or doctrinal authority. Intellectualized religious studies include not only the discussion of basic issues such as the nature, origin, form, function, and development law of religion, but also the investigation of various aspects of specific religious beliefs, organizations, rituals, culture, ethics, etc. As the founder of religious studies, Friedrich Max Müller abandoned the limitations of traditional European Christian theology and learned oriental languages, translated oriental religious classics, and used comparative religious methods to strive for an objective and fair study of rich and diverse religious phenomena. He edited and published a 51-volume Collection of Oriental Sacred Books, which included Buddhist scriptures and other religious documents from India, China, Japan, and other countries, providing important information for Westerners to understand oriental religions. Max Müller's groundbreaking research method not only laid the foundation for religious studies as a discipline, but also paved the way for Buddhism to enter the European world in modern times, thus forming a scholarly direction of the literature and philosophy in European Buddhist studies.

This academic style of interpreting the original meaning of oriental Buddhism with an academically neutral attitude has had a profound impact on the development of Buddhist studies in modern China. Especially in the process of translating Sanskrit Buddhist

scriptures into English, Western Buddhologists paid attention to etymological research on noun concepts and showed their understanding of Buddhist cultural characteristics through horizontal comparative studies of Buddhism and other religions. This point just inspired the intention of a group of laymen with profound Buddhist backgrounds in modern China. These people are not simple believers in Buddhism. Instead, they focused on the need for rational thinking and emphasized the importance of studying Buddhist scriptures, as they advocated the principle of "follow[ing] the scriptures rather than the authority 依法不依人", thus clearly distinguishing the true Dharma from its false counterpart. Such advocacy contrasts sharply with the anti-intellectual tradition formed in the Ming and Qing dynasties that tended to practice Chan in an unchecked, frivolous manner, refrain from studying theories, overlook the need to read texts, and "follow[ing] the authority rather than the scriptures 依人不依法 (Li 1995, p. 47)". Under the new trend of Buddhist studies development, these elite laymen played a key role in the development of modern Buddhist education. In the middle of the Qing dynasty, laymen including Wang Jin 汪縉 (1725–1792), Peng Shaosheng 彭紹升 (1740–1796), and others created the Jian Yang Academy 建陽書院, the first academy where laymen could also deliver lectures on Buddhist doctrines, which, unexpectedly, set the precedent of allowing laymen to teach Dharma in public. Following the example of the Jian Yang Academy, many similar institutes were established, such as the Dharmalaksana Academy (法相學社Faxiang Xueshe) established by Fan Gunong 範古農 (1881–1951) in Shanghai, the Yogacara Society (瑜伽學會 Yujia Xuehui) set up by Gu Jingyuan 顧淨緣 (1889–1937) in Shanghai, the Dharmalaksan Research Institute (法相研究會 Faxiang Yanjiuhui) and the Three Times Society (三時學會Sanshi Xuehui) created by Han Qingjing 韓清淨 (1884–1949) and others in Beijing, the Learning and Practicing Vihara (解行精舍 Jiexing Jingshe) founded by Wang Hongyuan 王弘願 (1876–1937) in Guangzhou, the Lotus Vihara (蓮花精舍 Lianhua Jingshe) organized by Wang Jiaqi 王家其 in Kunming and the Vimalakirti Vihara (維摩精舍 Weimo Jingshe) started by Yuan Huanxian 袁煥仙 (1887–1966), Jia Titao 賈題韜 (1909–1995) and others in Chengdu (Wang and Wang 2013, p. 101).

The establishment of modern Buddhist institutes began with the work of Yang Wenhui 楊文會 (1837–1911), a lay devotee living during the years of the late Qing. As early as the 1870s and 1880s, Yang Wenhui realized that, without cultivating new talent in the Buddhist community, Chinese Buddhism would not only face competition from other foreign religions but would also suffer suppression by the domestic ruling powers. After more than 10 years of hard preparatory work, the Jetavana Hermitage (祇洹精舍 Zhiyuan Jingshe) was finally inaugurated in 1908, the 34th year of the Guangxu Era. Basing itself on the Jinling Sutra Printing House (金陵刻經處 Jinling Kejing Chu), the Jetavana Hermitage attracted many highly recognized scholars to come and organize talks and discussions and, in this manner, created a good academic atmosphere. The case of the Jetavana Hermitage clearly demonstrates the combination of modern Buddhist studies and new educational practices. Thanks to its excellent academic background, high-quality teachers, and many other advantages, the Jetavana Hermitage showed a high level of teaching effectiveness and produced a generation of Buddhist elites such as Ouyang Jingwu 歐陽竟無 (1871–1943), Mei Guangxi 梅光羲 (1880–1947), Gui Bohua 桂伯華 (1861–1915), Li Zhenggang 李政剛, as well as Taixu 太虛 (1890–1947), Renshan 仁山 (1887–1951) and so on, turning the place into the most important Buddhist cultural center where Buddhist talents gathered (He 1998, p. 117). Comparing the Jetavana Hermitage with the self-organized Buddhist institutes and monastic education associations that were common at the time, one can see that, in terms of the concept of Buddhist education, the Jetavana Hermitage showed many signs of progressiveness. In contrast to the passive attitude adopted by many Buddhist practitioners, whose efforts to organize Buddhist institutes were driven merely by the wish to save monastic properties, Yang Wenhui made it clear that his work aimed at "the making of Buddhist teachers". Although the Jetavana Hermitage existed for less than two years, it had far-reaching significance. As it established an education system centered on research and investigation, the Jetavana Hermitage managed to break

free from the constraints imposed by the traditional Conglin Style education and create an independent educational space, the organization and institutionalization of which highly resembled those of modern schools. Notably, "aside from its application of innovative pedagogic methods and its focus on the teaching of canonical Buddhist texts, it was also the first Buddhist institute to teach English and Sanskrit language courses, which set an example for other Buddhist institutes established after it (Yu 1995, p. 318)."

Following the example set by Yang Wenhui, Ouyang Jingwu then established the Chinese Metaphysical Institute in Nanjing in 1922, the 11th year of the Republican era. As its educational principles, the institute "mourns the death of the true Dharma and dedicates itself to learning from the West 哀正法滅，立西域學宗旨", while at the same time, it "shows compassion for all those who suffer and works toward the common good of all people 悲眾生苦，立為人學宗旨" Meanwhile, the starting point of its education was to "open students' minds and cultivate students' interest in reading Buddhist texts through teaching 教授以誘進閱藏，開啟心思為鵠的 (*Inner Studies. No.3 Teaching Notes: The University Secto 1926*)." Based on this aspiration, the institute dedicated itself to training Buddhist specialists by combining the spirit of great compassion in Mahayana Buddhism with the spirit of patriotism born after the May Fourth Movement, as it insisted that "compassion be put before learning 悲而後有學" and "saving the nation be put before learning 救亡圖存而有學." Moreover, the institute required students to "pursue studies to benefit others 為利他而學" and to switch their aim from "entering the spiritual world" to "making positive contributions to the secular world". In this manner, the institute combined the tasks of revitalizing Buddhism and saving the Chinese nation closely with the civic awareness required of modern Chinese citizens following the establishment of the Republic of China. Thanks to its excellent academic atmosphere, the Chinese Metaphysical Institute produced a generation of outstanding Buddhist scholars such as Lü Cheng 呂澂 (1896–1989), Tang Yongtong 湯用彤 (1893–1964), and Xiong Shili 熊十力 (1885–1968). It also attracted more than 200 researchers to conduct research at the institute, along with thousands of students who attended to pursue their studies (Deng 1999, p. 18).

Through the intellectualization of faith, the laymen managed to shift people's focus to the rational components within Buddhist doctrines. Specifically, they promoted Buddhism as a form of knowledge in harmony with the spirit of modern science and, in this manner, facilitated the gradual intellectualization and rationalization of the Buddhist faith, which was often criticized as being tantamount to superstition. The laymen's efforts also had far-reaching implications for the later development of Buddhist studies and research (Yao 2013, p. 53). Additionally, those Buddhist research spaces created by knowledgeable laymen were in line with the contemporary pursuit of scientific rationality and speculation; they paved the way for the later transition of monastic education from its traditional Conglin Style to a rational, systematic modern model of Buddhist studies focusing on research and investigation. The remarkable contributions of laymen to Buddhist studies subsequently earned them a voice in the Buddhist community, and their influence on the development of Chinese Buddhism in the modern era was gained precisely from their foresight regarding knowledge. The laymen of modern China had played such a vital role in the promotion of Buddhist knowledge that, when recalling Buddhist research during the Late Qing, one can scarcely feel the participation of monastics. As Zhang Taiyan observes, "Since the Qing dynasty, Dharma has left those wearing Kasaya to be with the senior laymen 自清之季，佛法不在緇衣，而流入居士長者間."

### 2.2. Laymen's Efforts to Preserve the Space of Chinese Buddhist Education

In addition to the elite laymen in academia, those in other social sectors also played a key part in resisting the expropriation of monastic properties, funding the establishment of Buddhist institutes, and facilitating the publication of Buddhist journals and magazines. In the military sector, the lay devotee Lin Sen 林森 (1868–1943), president of the Nationalist government, was also a vegetarian and a devoted follower of Buddhism. Together with Taixu and others, Lin took the initiative to build a depository of Buddhist sutras in

front of Dr. Sun Yat-Sen's Mausoleum in Nanjing. Lin also photocopied 15 volumes of the Dragon-King sutra and ordered related government agencies to protect Qixia Temple and its properties, which in this manner contributed to resisting the movement of expropriating monastic properties. Meanwhile, when serving as the local governor and the commander-in-chief of China-Eastern Railway, lay devotee Zhu Ziqiao 朱子橋 (1974–1941) helped Taixu set up Buddhist institutes and revitalize Buddhism in Northeast China. During the Anti-Japanese War, Zhu made large contributions to Buddhism's revitalization in Northwest China, where he committed himself to renovating pagodas and establishing Buddhist institutes. Zhu also created the Ci En Academy (慈恩學院) and photocopied various Buddhist scriptures, including the Golden Canon of Zhaocheng (趙城金藏 Zhaocheng Jinzang) (Fori 1998, p. 15). Then, in the business sector, Wang Senfu王森甫, a very wealthy merchant from Wuhan, and Yu Huiguan 玉慧觀 (1891–1933), the owner of a pharmaceutical company based in Shanghai, had both become disciples of Taixu and subsequently provided tremendous financial support to facilitate the latter's activities to promote Buddhist education in Wuhan and Shanghai. Additionally, Wang Yiting 王一亭 (1867–1938), a lay devotee from Shanghai, who had acted as the director of the China Jisheng Society, the president of the World Buddhist Lay Association, and the chairman of the Shanghai Buddhist Bookstore (上海佛學書局 Shanghai Foxue Shuju), had made significant contributions to the development of Buddhist education throughout his life. Finally, the two brothers Jian Zhaonan 簡照南 (1870–1923) and Jian Yujie 簡玉階 (1875–1957), who were recognized entrepreneurs, donated their residence, the South Graden (南園 Nanyuan), to the Buddhist community in Shanghai, turning the place into a major site where the Shanghai Buddhist Pure Karma Society (上海佛教淨業社 Shanghai Fojiao Jingye She) and the Shanghai Buddhist Laymen Association (上海佛教居士林 Shanghai Fojiao Jushilin) could carry out their activities to promote Buddhism (Wang and Wang 2013, p. 102).

Due to their wealth, in their efforts to facilitate the development of Buddhist education, elite laymen were often capable of securing a strong economic base for circulating Buddhist doctrines. For this reason, in the modern era, many senior monastics were willing to closely collaborate with the lay community. Subsequently, the monastics walked out of the temples to dedicate themselves to the development of Buddhist education together with laymen. In fact, many Buddhist institutes were jointly organized by monastics and laymen. For example, the Hua Yan University (華嚴大學 Huayan Daxue), founded in 1914, the third year of the Republican era, within the Hardoon Garden (哈同花園 Hatong Huayuan) in Shanghai, was precisely an outcome of the collaboration between Zongyang 宗仰 (1865–1921), Yuexia 月霞 (1858–1917), and the owner of Hardoon Garden, also the largest property developer in Shanghai at the time, Silas Hardoon and his wife Luo Jialing 羅迦陵 (1864–1941). The Hua Yan University was a modern religious university that took Huayan Buddhism as its main teaching guide. It was also the first modern Buddhist institution of higher education that was ever known in Chinese history as a "university". Following Zongyang and Yuexia, monastics such as Dixian 諦閒 (1858–1932), Xingci 興慈 (1881–1950), Taixu, and Yuanying 圓瑛 (1878–1953) also maintained a close relationship with the lay community. In this manner, monastics and laymen worked together to advance the development of modern Chinese Buddhist education.

*2.3. The Improved Social Status of Laymen and the Changing Power Relations between Monastics and Householder Practitioners*

The changes that had occurred in relation to laymen's social status in modern China also illustrate the internal structural changes of modern Chinese Buddhism. The elevation of laymen's social status and the deterioration of that of monastics modified people's long-held belief in the superiority of the latter and the inferiority of the former. Since Buddhism was introduced to China over 2000 years ago, over time, it has created a set of systems aimed at securing the absolutely dominant position of the monastic community over the laymen. In Buddhist traditions formed in ancient China, monks/nuns were the true followers and advocates of Buddhism, whereas laymen, or household practitioners, could be its

only external defenders (Li 1993, p. 7). Such traditions that valued monastics over their lay counterparts were then preserved and passed down through the Chinese Buddhist education system and the practices of Dharma transmission. Specifically, given the dominant position of monastics, laymen were required to show them due respect and not criticize or judge their decisions or behavior. Consequently, a householder practitioner should be "as careful to serve a monastic as a servant was to serve his/her master 膽應奉事唯謹, 一如奴僕之事主人 (Lan 1997)". Laymen were not only prohibited from setting up altars to teach Dharma but also from creating Buddhist associations outside the temple or accepting disciples privately, which largely indicates the monastics' near monopoly of Buddhist education. In this regard, one can say that, in ancient China, Buddhist education was dominated by one group only: the monastics.

However, these seemingly solid power relations were to change in the context of modern China. The social stratification in modern Chinese cities, people's increased economic mobility, the development of new communication technology, the accelerated pace of life, and other societal changes had all modified how monastics and laypeople interacted with each other. On the one hand, the differences between their identities and the division of their related rights and responsibilities became increasingly institutionalized. On the other hand, the separation between Buddhist followers and nonfollowers was further institutionalized, thus making the lay community an integral part of the Chinese Buddhist Community on an institutional level (Ji 2014, p. 86). In the modern era, one can find the presence of elite laymen in many emerging fields such as academic research, business and commerce, new technology, media, and communications. Additionally, there were many politicians in the lay community. Due to the wealth of resources at their disposal, these elite laymen were able to facilitate the development of Chinese Buddhism in many ways. In modern China, the lay community took the initiative to respond to the needs of the revolution by seeking inspiration from Buddhist doctrines. At the same time, laymen participated in Buddhist studies and played a leading role in the development of monastic education and the organization of modern Buddhist institutes. In this regard, after the monastics, they formed another major body for Buddhist education and became a leading force in the revitalization of Chinese Buddhism. The awareness of their importance in political, economic, and social life also prompted laymen to adopt new strategies to challenge the traditional power relations between the lay community and the monastics. Just as the structure of knowledge production and power discourse in modern education systems are aligned, the Buddhist knowledge in the Buddhist system represents the intellectual virtue, the orthodoxy of Buddhist lineage, and the symbolic power that enable this mode of lay teachers teaching monks to break through the taboo of monastic education that monks should not rely on "white clothes白衣" to learn the Dharma. For instance, cultural elites such as Ouyang Jingwu attempted to loosen the restriction imposed by the norm that "only monastics are allowed to become masters, only renunciants are allowed to become monks/nuns 非僧不許為師，非出家不許為僧" through the discourse of Buddhist studies. Although this attempt was unsuccessful, such efforts themselves signaled the lay community's ability to challenge the status of monastics as the embodiment of Dharma, along with their moral privileges, and to organize itself into an independent social group in the modern era.

## 3. The Establishment of "Intellectualized" Buddhist Institutes via the Collaboration of the Monastic and Lay Communities

*3.1. The Traditional Conglin System and Its Hindrance to Establishing a Modern Buddhist Epistemology*

Due to the influence of patriarchy, the traditional Conglin system developed a strong belief in family bonds and ownership. This belief made it extremely difficult to break or change the traditional Dharma transmission systems, which were formed either based on the Buddhist schools followed or the tonsure ceremony performed, and the system of privatizing monastic properties, to start to promote modern educational models in Buddhist temples. The reason for this difficulty is that the tensions between rational, scientific views



and the traditional educational philosophy would inevitably undermine the authority of the Conglin educational model and, in this manner, harm the elders' interests, who were the resolute upholders of traditional values. Hence, at the beginning of establishing the Buddhist institutes, members of the Buddhist community who were supporters of the modern education system suffered considerably, as their work offended the interests of the established system. For instance, in 1904, the 30th year of the Guangxu Era, the Buddhist institute jointly created by Jing'an 敬安 (1851–1912) and Songfeng 松風 was loathed by local conservative monastics in the Hangzhou area, which eventually led to the tragic death of Song Feng. Later, Jing An wrote a poem commemorating this event. The poem reads: "In the end of the world we together with the desire to reverse the situation. Did you ever expect that you would end up sacrificing your life for the sake of Dharma? It is wailful that blood must be shed to make changes! You will certainly be remembered as the Buddhist who started the new era! 末劫同塵轉願運，那知為法竟亡身？可憐流血開風氣!師是僧中第一人！". Another tragedy took place in 1906, the 32nd year of the Guangxu Era. Shortly after Wenxi 文希 set up a Buddhist middle school in Tianning Temple, Yangzhou, he was groundlessly accused of maintaining secret connections with Japanese revolutionaries who were seeking refuge in China. He was then arrested and sentenced to lifetime imprisonment, and the middle school he established was also forced to close. These examples are clear evidence of the difficulties encountered by Buddhist practitioners in the early days as they tried to establish Buddhist institutes.

Taixu also experienced many ups and downs during the process as he tried to introduce reforms to Buddhist education, which demonstrates the extent to which the conservative sector of Chinese Buddhism resisted new educational concepts and practices. From another angle, Taixu's experience shows the determination of the new generation of Buddhist practitioners to advance Chinese Buddhist education. In 1912, the first year of the Republican era, Taixu and Renshan "made a big fuss" in Jinshan 金山 in their efforts to create a Buddhist university. Their attempts were ultimately unsuccessful due to opposition from the conservative sector of the Buddhist community, which made them realize the difficulties they would face should they wish to introduce new educational models within the existing Conglin system. Despite this unsuccessful attempt, Taixu did not abandon his hopes for promoting modern Buddhist institutes under the constraints imposed by the Conglin system. Thus, in 1917, the sixth year of the Republican era, when carrying out reforms at Jingci Temple 淨慈寺, once again, Taixu tried to "create the Yong Ming Vihara for the purpose of promoting Buddhist studies and cultivating Buddhist practitioners 籌設永明精舍，以作研究佛學，栽培弘法人材的地方." Nevertheless, the reforms were again met with objections from conservative elders in the Buddhist community of Hangzhou. "Due to their bad habits, the retired elders and senior monks in the temple, who were unwilling to follow new rules, established secret connections with the local gentry and military, as well as monks from other temples 寺中囿於惡習不甘拘束的退居與老班首等，勾結諸山寺僧及豪紳軍人." Together, they launched groundless and severe criticisms against Taixu and ultimately forced him to leave Jingci Temple. As with Taixu's unsuccessful attempt to establish a Buddhist university in Jinshan 10 years prior, the failure of Taixu's reforms at Jingci Temple is evidence of the difficulty of changing the conservative views held among certain sectors of the Buddhist community at the time and promoting modern Buddhist education under the constraints of the traditional Conglin system, a goal whose achievement seemed extremely unlikely.

Given the obstacles encountered by members within the Buddhist community as they tried to install a modern Buddhist education system, people in the education sector, who were better informed of the international situation, started to realize that the successful introduction of modern Buddhist education to China could be achieved only by avoiding all the restrictions and constraints imposed by the traditional Conglin system. In other words, ways must be found to organize modern Buddhist institutes outside the Conglin system. The new era thus also introduced new requirements for the proponents of Buddhist education. They were not only expected to demonstrate a good understanding of

Buddhist doctrines and the Buddhist faith but also needed to be capable of grasping the trends in the modern world and showing an in-depth understanding or firsthand experience of the status and values of new ideas. Only by fulfilling such requirements would they be able to observe Chinese society and adapt to this society from the standpoint of Buddhism (Deng 1999, p. 14).

### 3.2. The Integration of New Buddhist Institutes into Modern Education

The secular society emphasized the importance of disciplinary knowledge. Such knowledge, in turn, was evidence of the discursive power of "science". Thus, should it wish to modify its superstitious and backward image, Buddhism needed to "base itself on science to establish the highest faith from a scientific perspective (Huang 1995, p. 53)" and integrate rational, modern values into Buddhist education. The transition from Conglin Style education to institute-based Buddhist education is a key turning point in the history of Chinese Buddhism (Dongchu 1974, p. 204). The new education was aimed not only at training religious preachers, but also (and especially) at cultivating loyalty to the government and respect for the political ideologies of the Republic (Travagnin 2017, p. 230). With the subsequent development of modern Buddhist education, the term "Buddhist institute" proposed by Taixu became the most commonly used term to refer to Buddhist educational institutions (Zhang 2014, p. 216).

In 1918, the seventh year of the Republican era, with the support of Zhang Taiyan 章太炎 (1869–1936), Chen Yuanba i 陳元白, Wang Yiting 王一亭, Jiang Zuobin 蔣作賓, and others, Taixu founded the Awakening Society (覺社 Jue She) in Shanghai, which inspired Taixu to develop other educational ideas such as the introduction of a university sector within Buddhist institutes, the creation of scripture perusal chambers, lecture halls, and publishers specializing in the publication of Buddhist works. Then, on 1 September 1922, in the 11th year of the Republican era, Taixu founded the Wuchang Buddhist Academy (武昌佛學院 Wuchang Foxueyuan) on Qianjia Street (千家街 Qianjiajie) within Wang Shan Men 望山門 of Wuchang City. This event had far-reaching significance in the history of modern Buddhist education, as it was through this academy that Taixu's hypothesis of the revitalization of Chinese Buddhism and the cultivation of new Buddhist experts was tested. Taixu's educational philosophy can be synthesized from his advocacy for revolution in the domains of Buddhist doctrines, Buddhist systems, and Buddhist property, which clearly shows the extent to which Taixu's educational thought was informed by his thorough reflection on the modern transition of Chinese Buddhism. Following the Wuchang Buddhist Academy, Taixu founded several other new Buddhist Academies whose influence is also noteworthy. These included the Sino-Tibetan Buddhist Academy (漢藏教理院 Hanzang Jiaoliyuan), the Minnan Buddhist Academy (閩南佛學院 Minnan Foxueyuan), the Bailin Buddhist Academy (柏林教理院 Bailin Jiaoliyuan), and others, which trained a great number of modern Buddhist experts and significantly improved the quality of Chinese Buddhist practitioners in the modern era. In this regard, these modern Buddhist Academies had a profound impact on the modernization of Chinese Buddhism. Thanks to the efforts of progressive-minded educators in the Buddhist community, modern Buddhist educational institutions were set up all over the country. New Buddhist institutes were established even in the remote northeastern and northwestern regions of China (Deng 1999, p. 16). According to incomplete statistics, during the Republican years, there were approximately 157 Buddhist institutes in China, which spanned all 21 of the country's provinces at that time. Among these, Jiangsu Province and Zhejiang Province had the largest number of Buddhist institutes: 24 institutes were established in Jiangsu Province, and 14 were created in Zhejiang Province (Li 2009, p. 257). Between the 1920s and 1940s, Taixu and his disciples either established or taught regularly at 40 or 50 institutes at a minimum (Deng 1999, p. 9). In this manner, the appearance of the new Buddhist educational institutions broke the constraints imposed by the traditional Conglin system, facilitated the transition of the Buddhist educational model, and thus played a leading role in modernizing Chinese Buddhist education.

Compared to the traditional Conglin Style, the new Buddhist institutes had achieved many breakthroughs and introduced many innovations. In terms of their educational philosophy, the new institutes acquired the characteristics of social education. Moreover, instead of only teaching Buddhism, they taught subjects covering Western learning, Eastern learning, and even Christian theology. Meanwhile, the pedagogic methods employed were largely inspired by modern academic research, which clearly reflects the modernization and scientification of Chinese Buddhist education in the modern era. For example, it adopts scientific methods and theories to analyze and explain Buddhist phenomena; it introduces comparative and interdisciplinary perspectives to broaden the scope of Buddhist studies; it emphasizes the historical and social contexts of Buddhist texts and traditions; and it explores the practical implications and applications of Buddhist teachings for contemporary issues. In terms of teaching Buddhist doctrines, the new institutes favored the simultaneous study of multiple schools of Buddhism and their doctrines. Since the establishment of the Republic of China, some elders in the Conglin system have created Buddhist institutes to promote the doctrines of their own Buddhist schools. Examples include the Dharma Realm Academy 法界學院 (Fajie Xueyuan) of Changshu, the Hua Yan University of Shanghai, the Guanzong Academy 觀宗學社 (Guanzong Xuesh) of Ningbo, various institutes in Gaoyou, including the Tiantai Academy 天台學院 (Tiantai Xueyuan), and the many academies founded by Tanxu in northern China that belonged to the Tiantai Dharma Lineage. These Buddhist institutes tended to focus on promoting the doctrines of particular Buddhist schools and training monastics who were meant to become expertsin those schools. While specialized Buddhist education may facilitate the in-depth study of the canonical texts of specific Buddhist schools and, in this manner, further the development of those schools, this kind of Buddhist education can also nurture bias and factionalism among the different schools of Buddhism. In contrast, the new Buddhist institutes focused on Chinese Buddhism as a whole and encouraged the study of multiple Buddhist schools and their doctrines at the same time. For example, in the curriculum that Yang Wenhui designed for Jetavana Hermitage, "the Inner Class Curriculum of Buddhism (釋氏學堂內班課程 Shishi Xuetang Neiban Kecheng)", which was a fairly comprehensive curriculum, Yang included original Buddhist scriptures and various canonical texts from Mahayana Buddhism and Hinayana Buddhism. Yang's curriculum also stressed that "starting from the fourth year, students can decide, as they wish, to dedicate the next two, three or five years (or any length of time) of their life to study Buddhist scriptures. They may choose to study the scriptures of several Buddhist schools at the same time or to focus on those in a particular school, as they see fit 自第四年起，或兩年，或三、五年，不拘期限，各宗典籍，或專學一門，或兼學數門，均隨學人志願。". Then, in terms of the scope of the studies of Buddhist scriptures, the Chinese Metaphysical Institute, aiming at the revitalization of Indian Buddhism, especially the Nalanda model of Buddhist education, tried to include texts used in various Buddhist schools in its curriculum, including Mahayana Buddhism, Hinayana Buddhism, Madhyamaka Buddhism, Yogacara Buddhism, Esoteric Buddhism, and Exoteric Buddhism. In doing so, the Chinese Metaphysical Institute "hoped to project an image of Buddhism as a unity." Similarly, the Wuchang Buddhist Academy introduced pedagogic methods unlimited by the prioritization of single Buddhist schools. The course outline shows that, at the Wuchang Buddhist Academy, "scriptures from all Buddhist schools were taught". In 1925, the 14th year of the Republican era, Taixu further proposed that "new Buddhist universities should not emphasize the division of Buddhist schools." "Of the two approaches of organizing Buddhist institutes, one tends to encourage the institutions' specialization in particular Buddhist schools, whereas the other approach takes as its objective the revitalization of all Buddhist sects, the first approach often prioritizes the teachings of one specific Buddhist school without allowing students the opportunity to gain a balanced and comprehensive understanding of other Buddhist schools. The second approach, instead, enables students to study both Mahayana Buddhism and Hinayana Buddhism and thus to achieve a comprehensive view of Buddhist studies. Based on this view, students can

decide, according to their interests, in which Buddhist school they wish to specialize. In this manner, the second approach improves educational efficiency without undermining the distinctive characteristics of each Buddhist sect. At the same time, it paves the way for collaboration between temples of different schools when these are built in the future 一則以專宏一家宗風為事業，一則以普遍整興各宗教為鵠的也。且分宗則偏注一家，不能對各宗普遍了達，平均發展。不分宗則大小乘既得全體研究，於佛學有全整之認識，再以性質所近，深造一宗，既屬事半功倍，且不失嚴分宗派，則將來建各宗寺，更有互相協調之利。Taixu's theory of the simultaneous promotion of all eight Buddhist sects in Buddhist education was based on his advocacy of the equal development of Buddhist schools and the elimination of biases and sectarianism in Chinese Buddhism. In "What Do I Think of the Existence of Different Schools of Buddhism?", Taixu writes: "The eight schools under the Great Vehicle are all equal in their status. They are also equal in their final goal, which is the attainment of Buddhahood. Their only difference is the methods that they each employ to achieve that goal 這大乘八宗，其境是平等的，其果都是以成佛為究竟，也是平等的，不過在行上，諸宗各有差別的施設。". Taixu's advocacy for the equal development of the eight Buddhist schools is thus unmistakable. According to Fu Yinglan, "In his theory, Master Taixu conceptualized the eight Buddhist sects as a unity in which each sect could maintain its distinctive features, but at the same time, its existence would also depend on the existence of other sects. Specifically, each sect could judge and criticize other Buddhist sects according to its own principles and doctrines. It could also posit itself above all other schools, turning these into a part of it. In this sense, beyond each particular school, there would be no Dharma. However, at the same time, the existence and development of each sect also relied on the existence of other sects: without other schools, the individual sect would also perish. This conceptualization clearly demonstrates the equal status of the eight Buddhist schools under the Great Vehicle, without discarding the distinctive features of each school (Fu 2010, p. 204)."

In addition to the educational model that encouraged the simultaneous study of multiple schools of Buddhism, when designing the curriculum, the progressive-minded Buddhist educators also actively learned from the experiences and lessons gained in national and international religious and nonreligious education to accelerate the scientification and rationalization of modern Buddhist education. The "objective" and "scientific" approach to studying Buddhist doctrines, a product of the modern era, was key to adapting Buddhism to modern society and revitalizing Buddhist education. Therefore, the new Buddhist institutes introduced tiered learning, an advanced pedagogic method, and other critical research methodologies. For instance, the curriculum of Jetavana Hermitage, established by Yang Wenhui, learned from the successful practices of Japanese Buddhist education and European Christian education, including Catholic education. It incorporated modern subjects such as foreign languages, Western studies, and reformist studies. It invited Su Manshu 蘇曼殊 (1884–1918) to teach English and Li Xiaotun 李曉暾 to teach Chinese in order to expand the students' perspectives and knowledge.

This curriculum reflects Yang's attempt to combine Buddhist education with academic publishing and research. Then, Taixu's Wuchang Buddhist Academy, aiming at "the creation of a new form of Buddhism in line with modern thought by critically studying the current and past academic achievements of the East and the West," offered both intensive and sessional courses, thus adapting its curriculum to the requirements of modern teaching and education systems. The intensive courses were reserved for dedicated learning programs that usually lasted for three years, whereas the duration of sessional courses was only six months. In 1924, the 13th year of the Republican era, the intensive-course sector of the Academy was turned into a university sector that focused on academic research as much as teaching. The university sector of Wuchang Buddhist Academy had acquired clear features of modern Buddhism, as it combined the thoughts of different Buddhist schools and both metaphysical and physical studies. Meanwhile, apart from the teaching of the doctrines and origins of all Buddhist schools, courses taught at the university sector included Buddhist logico-epistemology, the history of Chinese Buddhism,

the history of Indian Buddhism, Chinese and Western philosophy, Western ethics, psychology, religious studies, sociology, biology, Sanskrit, the Tibetan language, English, and Japanese, among others. In terms of the teaching staff, apart from Buddhist scholars, the Academy also recruited university academics outside the Buddhist community, forming a staff team comprising both followers and nonfollowers. Additionally, the Academy created the Akarawathi Saddha Publishing House (正信印書館 Zhengxin Yinshuguan) and the magazine *The Sound of Sea Tide* (海潮音 Haichao Yin), which not only contributed to promoting the Academy but also offered staff and students chances to publish their research. Last, Ouyang Jingwu's Chinese Metaphysical Institute must also be mentioned. The Chinese Metaphysical Institute was divided into four sectors: the high school sector, the undergraduate sector, the postgraduate sector, and the travel-based learning sector. Courses taught in the undergraduate sector were also separated into four categories that included cram courses, preparatory courses, special courses, and undergraduate courses. In the high school sector, approximately one-third of the classes were dedicated to self-cultivation and Buddhist studies, while the remainder were reserved for subjects such as Chinese, English, history, geography, and the natural sciences. The undergraduate and postgraduate sectors focused instead on Yogācāra School while also teaching subjects that included the doctrines of Buddhist schools, Buddhist logic, Buddhist monastic discipline, Buddhist psychology, Buddhist art, Buddhist history, Chinese and Western philosophy, old Chinese, Sanskrit, Tibetan, English, Japanese, and so on. Notably, the Institute's undergraduate and postgraduate sectors embraced international academic standards by encouraging the use of presentations, discussions, and critical research as the primary methods for delivering course content, which was a radical break from the traditional educational model based on force-feeding knowledge (Deng 1999, p. 17) as well as the Conglin Style that tended to value morality over wisdom.

Compared to conservative monastics, progressive Buddhist educators were often more open-minded. During his creation of the Jetavana Hermitage and the Buddhist Studies Association, Yang Wenhui took care to place Buddhist studies in an international context and subsequently included Japanese, English, and Sanskrit studies in the curriculum of the Jetavana Hermitage. Yang's international vision of Buddhist education had a significant impact on Taixu and Ouyang Jingwu and their later organization of modern Buddhist institutes. In 1929, the 18th year of the Republican era, Taixu, who had just returned from his world trip, began to put forward a plan for creating the World Buddhist Academy and founded its "head institute" in Nanjing. Shortly afterward, Taixu established the Sino-Tibetan Buddhist Academy in Sichuan Province, making it the Sino-Tibetan sector of the World Buddhist Academy. He also created the "Library of the World Buddhist Academy", based in Wuchang Buddhist Academy, and turned the Minnan Buddhist Academy and the Bailin Institute of Beijing into the Sino-Japanese and the Sino-English sectors of the World Buddhist Academy, before establishing a Balinese sector of the Academy in Xi'an Province (He 2018, p. 160). Later in 1939, the 28th year of the Republican era, Taixu led a mission to Burma, India, Nepal, British Ceylon, Thailand, Singapore, Malaysia, and Vietnam to promote Buddhism and China's cause against the Japanese invasion. During his trip, he met with world-renowned leaders such as Nehru, Gandhi, and Tagore. Taixu also sent young Buddhists to study in Japan and South Asian countries, among whom one may recognize the names of Dayong 大勇, Chisong 持松, Xianyin 顯蔭, Manshu, Mochan 墨禪, Tanxuan 談玄, Tianhui 天慧, and Renxing 仁性, who went to study in Japan, and those of Tican 體參, Fafang 法舫, Baihui 白慧, and Xiulu 岫廬, who went to India. Others, such as Weihuan 惟幻, Fazhou 法周, Huisong 慧松, Weishi 唯實 and Liaocan 了參, went to study in British Ceylon. There were still other young Buddhists who went to study in Thailand and Burma, such as Dengci 等慈, Beiguan 悲觀, Shangui 善歸, Xingjiao 性教, Jueyuan 覺圓, Daju 達居, Jingshan 淨善, Changhai 昌海, and Chengru 誠如. All these Buddhists, who had studied overseas, would later become leading forces in promoting modern Buddhism both within and outside China.

Based on the Buddhist educational theories and practices of Yang Wenhui, Ouyang Jingwu, Taixu, and others, Buddhist education and the establishment of new Buddhist institutes at that time aimed at not only attaining wisdom to gain personal freedom but also contributing a "source of ideas" to facilitate social development. The new era thus brought new approaches to knowledge production. Meanwhile, in modern China, the reliance on written texts to transmit knowledge, the importance attributed to the establishment of new epistemologies, the evaluation of personal competencies, and even the design of the teaching space all started to build their own unique logic. By introducing the epistemologies, teaching systems, values, and beliefs celebrated in modern education into Buddhism, the Buddhist educators managed to redefine the content, forms, and subjects of the Buddhist legacy and changed how knowledge, power, and the sanctity of Chinese Buddhism were aligned (Ji 2009, p. 41).

### 3.3. The Learned "Student-Monks (学僧 Xueseng)" and the New Dharma Lineages

At the same time as Buddhist institutes were where all Buddhism-related knowledge was taught, they also formed a site for power struggles. In addition to the building of Buddhist networks based on Dharma transmission to defend the orthodoxy of each Buddhist lineage, the modern era saw the emergence of another method of asserting Buddhist orthodoxies, which was closely linked to the work of the " student-monks" in Buddhist institutes (Rongdao 2017, pp. 55–70). Notably, the functions of Buddhist institutes were by no means limited to those of teaching Buddhist knowledge. Instead, such institutes were also responsible for helping the monastic community keep pace with formal national education. The establishment of modern Buddhist educational institutions was aimed precisely at cultivating Buddhist experts with multidisciplinary skills who could adapt to the new era and contribute to the new society. The many " student-monks" who had graduated from these modern institutions subsequently modified the relationship between the Buddhist community and the Chinese nation, which was undergoing drastic social changes. As they learned secular knowledge and built a strong knowledge base, the student-monks were capable of exchanging Buddhist ideas with intellectuals and elites in the fields of religion, philosophy, science, etc. In this process, they were in fact defending the field of Buddhism. As Jichen 寄尘 suggests, "to introduce new reforms into Chinese Buddhism, one should not only examine the current social trends but also study the modern society thoroughly to understand how it is organized and what convenient methods can be applied in order to cultivate the new generation of Buddhist followers in the future!" He further points out that the combination of Buddhist education with social education can at least enable the monastics to "first, understand the way in which the modern society is organized, and second, to acknowledge the role played by Buddhism in the modern society". Keenly aware of their social and religious responsibilities, the student-monks who trained in modern Buddhist institutes were seen as active contributors to redefining Chinese Buddhism in the modern era and were optimistic about the part they would play in reshaping the Buddhist religion in the nation's future. As the social values of the student monks were continuously recognized by the Buddhist community in modern China, in the Jiangsu and Zhejiang regions, even abbots in the Conglin system would accept student monks as their disciples. It was precisely the tensions and conflicts between the traditional and the modern models of Dharma transmission that subsequently prompted the modernization of Chinese Buddhism. The new student monks should be regarded as an important force shaping modern Chinese Buddhism.

By now, it is clear that the transfer from civil society to the government as the main body for organizing Buddhist education has relied largely on the work of Buddhist experts cultivated by the modern Buddhist institutes, with Taixu being the most prominent figure in this regard (Chen 2020, p. 7). While his modern view of Buddhist education was still nascent during the Republican era, by 1956 to 1966, after establishing the People's Republic of China, the influence of his view had been fully felt. Of the teaching staff at the Buddhist Academy of China, aside from laymen and university lecturers, monas-

tic staff members such as Fazun 法尊, Guankong 觀空, Zhengguo 正果, Chenkong 塵空, Yejun 葉均, and Yuyu 虞愚 had all been either students or teachers at the Wuchang Buddhist Academy and the Sino-Tibetan Buddhist Academy founded by Taixu. Meanwhile, Zhao Puchu 趙樸初, Juzan 巨贊, and Mingzhen 明真 were all followers of Taixu's modern philosophy of Humanistic Buddhism (人間佛教 Renjian Fojiao), thanks to whose efforts the privileged position of Humanistic Buddhism in Buddhist studies today is widely recognized.

## 4. Conclusions

This paper focuses on "knowledge" as a discursive construction in modern Chinese Buddhism, and draws on the analytical framework of "discourse and power" by the French sociologist Foucault. By describing the evolution of Chinese Buddhist education from Conglin to Buddhist Academies, it shows the transformation of the social image and function of modern Buddhism from faith to knowledge culture, discusses why modern Buddhism needs to base itself on "knowledge" to connect the sacredness of Buddhism with the world, and what role the Buddhist monastic space as a "Buddhist Academy" plays in the new discursive practice.

In understanding the term "Buddhist education", this paper uses an outside-in approach, i.e., defining Buddhist education in light of the new social attitudes toward the concept, methods, and content of education at the time. Therefore, the concept of "Buddhist education" in the paper is not fixed, but changes with the changes in the external environment. It does not specifically refer to the forms of ancient Conglin such as master-disciple teaching, sitting incense in the meditation hall, preaching in the Dharma hall, and traveling around to visit eminent monks. After the introduction of Western learning in modern times, it broke the internal, enlightenment-oriented, and inspirational education of the monks, which focused on spiritual and mental quality. It reshaped people's thinking concepts with the modern knowledge system, which emphasizes science, rationality, empiricism, and discipline classification. Therefore, it changed the traditional Conglin training mode of education and also caused the Buddhist education venues to change from monastic communities to academies (including research organizations). When discussing the Conglin system, this paper also focuses on explaining the impact of "knowledge" as a discourse power on the traditional Conglin power structure. After the definition of education changed from introspective spiritual enlightenment to argumentative knowledge accumulation, "knowledge" as a new influence weakened the stability of the Dharma-clan relationship in the Conglin system. The stability of the Dharma-clan relationship in the traditional Conglin was also based on the Chinese social clan structure and produced at the same frequency. This paper argues that the development of Buddhism has always been "in tune with the social environment" in order to be viable.

The changes in the concept of Buddhist education in modern China have offered an important angle for observing and reliving the interactions and conflicts between Western and Chinese cultures in the modern era. Cultural differences between Westerners and Easterners have led to their different ways of thinking. The Chinese traditional private schooling system, which valued personal wisdom, intuition, and individuality, was completely different from the Western education system, which tended to be more practical, emphasizing the rational, tiered learning of academic subjects. Therefore, during the eastward transmission of Western learning in the modern era, a period when China was undergoing deep social changes, the absorption of Western knowledge, the need to cultivate "new citizens", and changes in the national views of education all led to the transformation of the Conglin education system of Chinese Buddhism, which was deeply affected by Confucianism. This transformation, in turn, involved many complex issues, including the traditional Conglin patriarchy system, the Conglin education system, the methods of Dharma transmission, and the relationship between monastics and householder practitioners, among others.

Revisiting the practices of modern Chinese Buddhist education reveals several prominent features in the development of Chinese Buddhist education in the modern era. Such

features include the shift of Buddhism's passive and conservative view on education to a more active and open view; the gradual abandonment of the traditional Conglin educational model and the embrace of the modern, institutionalized model; and the end of the monastics' monopoly of Buddhist education and the beginning of the joint organization of Buddhist education between monastics and laymen. In this process of the overall rationalization of Chinese education, Buddhist education, as an important part of traditional Chinese education, was unavoidably affected. The rationalization of Chinese Buddhist education subsequently became a major topic in modern Buddhist reforms. Buddhist masters such as Yuexia, Dixian, Yang Wenhui, Tanxu, Changxing, Yuanying, and Taixu had all dedicated themselves actively to promoting modern Buddhist educational practices. With their awakened awareness of modernity and globalization, progressive-minded members of the Buddhist community attempted to integrate the Buddhist religion into world civilization through the development of Buddhist education. Subsequently, they tried to shift their focus from "a China-centered Buddhism to the creation of a new Buddhism more adaptable to the needs of our time 從中國漢族的佛教本位，而適合時代需要的新佛教." In this regard, Taixu's work is particularly noteworthy, as he promoted the notion of Humanistic Buddhism and traveled around Europe and America to engage in conversations with Western religious leaders from different schools. He also contributed actively to the creation of the World Buddhist Academy and the World Buddhist Library and, in this manner, facilitated the globalization of Buddhism. Such efforts not only enabled the religious culture of modern China to break free from the constraints imposed by ancient traditions and cultural borders but also directly impacted the promotion of Chinese religious culture, including its national characteristics and diverse development pathways, in the international world. Since then, Buddhism has widely participated in various cultural dialogues through various forms, actively participated in international exchanges and cooperation, broken the imprisonment of various regions and nationalities, and actively responded to various social and cultural trends of thought, thus moving towards the development path of traditional and modern, national and world-wide opportunities.

**Funding:** This research was funded by the National Social Science Fund of China, grant numbers 17ZDA233 and 21CZJ002, and the New Young Teachers' Research Start-up Project, Beijing Foreign Studies University, grant number 2021QD005.

**Institutional Review Board Statement:** Not applicable.

**Informed Consent Statement:** Not applicable.

**Data Availability Statement:** Not applicable.

**Conflicts of Interest:** The author declares no conflict of interest.

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
