# Peer review of "From “Sangha Forest” (叢林 Conglin) to “Buddhist Academy”: The Influence of Western Knowledge Paradigm on the Chinese Sangha Education in Modern Times"

_religions, doi:10.3390/rel14081068_

Round 1
Reviewer 1 Report
Please see attachment.

Author Response
I would like to express my sincere gratitude for your valuable comments on my paper. I have carefully addressed each of your review suggestions and made corresponding revisions to the paper. Please see the attachment for my responses.

Reviewer 2 Report
Also can be refer to below book:
Deng, Zimei, Chen, Bing 邓子美、陈兵, 2000, Chinese Buddhist in 20 Century 二十世纪中国佛教, Beijing, Minzu Publishing
Author Response
Thank you for your valuable feedback on my paper. I have revised it according to your suggestions.
I have added the reference book you recommended: Deng, Zimei, Chen, Bing (2000). Chinese Buddhist in 20 Century. Beijing: Minzu Publishing.
Please find the updated version of my paper attached to this email. I hope it meets your expectations and requirements.

Reviewer 3 Report
Comments on “From “Conglin(叢林)” to “College”
You have made the historical period you are working with clear, the end of the Qing into the Republican period. But I think it would help your argument if you make some attempt to describe education in general. The late Qing and early Republican period is very important because China would develop educational institutions in response to the introduction of Western institutions into China. I believe you need to recognize the way educational institutions entered China.
The changes to Buddhist education occur within a larger context that you indirectly refer to. I think you need to cover some more general and overarching changes that were occurring during this time.
You mention The Birth of the Prison by Fucault. You cite Foucault, but it seems like you are just putting a title by Foucault in without actually working with aspects of this text that could help your argument. The section that might help is “Docile Bodies,” “The control of activity,” pp. 149-169. Many people think Foucault was some kind of radical thinker, but his historical understanding of institutions is unparalleled in Western theory, and I think some of his ideas would help your argument. I would also recommend Ruth Hayhoe, China's Universities, 1895-1995: A Century of Cultural Conflict.
l. 56-61: “The perceived superiority of Western education over its Eastern counterpart in all respects then led intellectuals to reflect on Chinese traditions, culture and educational models. Following this critical reflection, educational traditions dominated by Confucianism were eventually replaced by new trends in science education and social education, while the discursive power of “science”, the newly instilled epistemology, increased continuously.”
You make some interesting points. But you do not seem to integrate these points into a form that would help your arguments.
l. 84: why not put Foucault’s book title in italics? The Birth of the Prison.
l. 187: 2. The Rise of Knowledgeful [Knowledgeable?] Lay Devotees
l. 286-289: “ . . . this elitist Buddhist research space paved the way for the later transition of monastic education from its traditional conglin model to a rational, systematic modern model of Buddhist studies focusing on research and investigation.”
So an elitist research space is good? You scan over hundreds of years of Buddhism to claim the “modern” Buddhist practices are better? This is not much of an argument.
l. 519-521: “Meanwhile, the pedagogic methods employed were largely inspired by modern academic research, which clearly reflects the modernization and scientification of Chinese Buddhist education in the modern era.”
Besides giving particular ideas about Buddhist scripture, you make statements like this which, unless you try to explain what you mean by “modernization” and “scientification” in the context of Buddhism. I think you should bring this out more in your paper because this seems to be the main idea you are trying to bring out for your readers.
l. 599-603: “For example, the curriculum developed by Qi Yuan Vihara, founded by Yang Wenhui, had learned from the success of Japanese Buddhist education and European Christian education, including Catholic education, as it adopted a modern educational system and included courses such as foreign languages, Western learning, subjects promoted during the Hundred Days’ Reform and so on.”
I think you need to emphasize these sorts of ideas. Obviously when you talk about the modern and scientific reform of Buddhism, you are speaking about the way Buddhism availed itself of modern approaches and structures of curriculum and pedagogy.
pp. 10-13: You make some good points here. I think you need to highlight the sheer number of colleges and universities. Obviously the government would have encouraged these changes? This is important. The Nationalists were authoritarian (just like the CCP would be after 1949), but he chose well when promoting certain aspects of social and institutional reform.
l. 730: You mention “Discursive practice,” which may be a method derived from Foucault, but in Buddhism discourse is also extremely important. I think you need to highlight more discourse that brings evidence for your argument. More discourse about education, la little less about Dharma.
& I find it very hard to believe Buddhism was in such a bad state before the moder period. From what I understand, Chan promoted lay people to come to Buddhism. Popular Buddhism is a very important innovation by Chan.
Author Response
Thank you for giving such a sincere and detailed feedback on my article. I know I have many shortcomings in my writing, so I have revised the whole text based on your suggestions. Please see the attachment and the edited version for my response. I appreciate your help very much!
Please find the updated version of my paper attached to this email. I hope it meets your expectations and requirements.

Reviewer 4 Report
Please see the attachment

Author Response
Thank you very much for your sincere evaluation of my paper. Please see the attachment for my reply. Please find the updated version of my paper attached to this email. I hope it meets your expectations and requirements.

Reviewer 5 Report
This article discusses the institutional reforms and educational content of Buddhist education at the beginning of modern China. The argumentation is solid and supported by rich materials, providing a relatively comprehensive analysis.
However, the article has certain issues as below:
(1), about the article's choice of theoretical perspective: Michel Foucault's theory of "space-power" seems not that relevant to this article's concern, removing this theory would not impact the overall argument. Furthermore, Foucault's theory defines "space" as seen in panoramic surveillance within prisons, which does not align with the spatiality of temples. Therefore, Foucault's concept of "space-power," with its distinct modernity critique, may not serve as a suitable theoretical framework for the Buddhist education discussed here.
(2) The author's use of "power" as a keyword could be a way to use Foucault’s theory. The concept of "power" challenges the traditional political philosophy's notion of "sovereignty" by highlighting its practice in various everyday relations, different education system can be seen as the platform for power.
While the article vividly portrays the transformation of Buddhist education from "conglin" to "college" through detailed examples, it fails to specify the underlying context of this process. It is possible that the profound changes in Buddhist education during modern China were not solely due to the incorporation of modern educational ideas. Rather, it may have resulted from Buddhist education liberating itself from strong political constraints and gaining relative freedom within the backdrop of modern China's lack of "sovereignty". Power in this way could be interwoven into the process of changes.
(3) The article excellently depicts the collaboration between monastics and laypeople in the field of education. The power and contributions of laymen were immense in the reform of modern Buddhist education, and many of them were active reformers and social activists. If the author contextualizes the background and explains why these individuals fervently promoted Buddhist education and whether certain Buddhist concepts aligned with the requirements of modern China's transformation, the discussion of "power" in Buddhist knowledge would be more consistent with the article's narrative logic.
Author Response

(The authors gave the same response as above.)
